TECHNICAL RELEASE

# Galaxy QCxMS for straightforward semi-empirical quantum mechanical EI-MS prediction

Wudmir Y. Rojas[1], Zargham Ahmad[1], Julia Jakiela[2], Helge Hecht[1,*], Jana Klánová[1] and Elliott J. Price[1]

1 Faculty of Science, Masaryk University, RECETOX, Kotlářská 2, 60200, Brno, Czech Republic
2 School of Chemistry, University of Edinburgh, Edinburgh, UK

## ABSTRACT

High-performance computing (HPC) environments are crucial for computational research, including quantum chemistry (QC), but pose challenges for non-expert users. Researchers with limited computational knowledge struggle to utilise domain-specific software and access mass spectra prediction for *in silico* annotation.

Here, we provide a robust workflow that leverages interoperable file formats for molecular structures to ensure integration across various QC tools. The quantum chemistry package for mass spectral predictions after electron ionization or collision-induced dissociation has been integrated into the Galaxy platform, enabling automated analysis of fragmentation mechanisms. The extended tight binding quantum chemistry package, chosen for its balance between accuracy and computational efficiency, provides molecular geometry optimisation. A Docker image encapsulates the necessary software stack.

We demonstrated the workflow for four molecules, highlighting the scalability and efficiency of our solution via runtime performance analysis. This work shows how non-HPC users can make these predictions effortlessly, using advanced computational tools without needing in-depth expertise.

**Subjects** Software and Workflows, Human and Biomedical Sciences, Metabolomics and Proteomics

**Submitted:** 21 October 2024

* Corresponding author. E-mail: helge.hecht@recetox.muni.cz

Preprint submitted at https://doi.org/10.26434/chemrxiv-2025-fd68z

## INTRODUCTION

The structural annotation of mass spectrometry (MS) data is a critical step in many scientific workflows [1]; however, it often presents a significant bottleneck. Confident annotation relies upon spectral library matching against references from analytical standards, but in many cases chemical standards are inaccessible [2]. Spectra predicted *in silico* from molecular structures can act as useful substitute records when reference spectra are lacking, and offer insights into fragmentation dynamics and reaction mechanisms [3, 4]. The specialised software required for quantum chemistry (QC)-based predictions, like the QCxMS framework (RRID:SCR_026928) [5, 6, 7], is often complex, resource-intensive, and demands substantial user training. High-performance computing (HPC) environments are essential for QC and other computationally intensive research fields. However, their complexity can be daunting for non-experts and is a barrier to the efficient use of domain-specific software for many researchers. For example, the QC-based prediction of mass spectra for *in silico* annotation is inaccessible to many wet-lab scientists lacking a background in HPC.

The Galaxy platform addresses these challenges by providing access to HPC resources via a user-friendly web interface and offering a suite of tools across various domains [8]. This is further supported by interactive, self-paced and user-focused training materials developed by the Galaxy community to improve accessibility [9, 10, 11]. Integrating the semi-empirical quantum mechanics (SQM) capabilities of QCxMS and xTB [12, 13] aims to democratise *in silico* mass spectrometry annotation within Galaxy. An end-to-end QCxMS workflow, built upon a previously published semi-automated HPC tool set [14], has been integrated into the Galaxy platform, following the Findability, Accessibility, Interoperability, Reusability (FAIR) principles for research software [15]. We demonstrated and evaluated this Galaxy workflow on four selected molecules with varying chemical and structural complexity as case studies. Lastly, we discuss accessibility and usability, particularly for researchers without computational expertise.

## IMPLEMENTATION

### Workflow

The Galaxy workflow shown in Figure 1 integrates several Galaxy tools, which are described in more detail in the following section on the Galaxy platform, to perform a comprehensive molecular simulation and optimisation process for the prediction of electron ionization (EI) mass spectra. The workflow is compatible with other chemo-informatics Galaxy tools from the *ChemicalToolbox* [16], and leverages standard data formats supported by Open Babel (RRID:SCR_014920) [17]. This allows researchers to use a SMILES code representing a molecule for generating the SDF required as the workflow input and obtain a mass spectrum in `msp` format, which can be directly used by annotation software. The implementation supports processing datasets with multiple molecules, leveraging Galaxy's collections framework for efficiency [18]. Each step retains only strictly required outputs for a lean memory footprint. Therefore, the workflow is suitable for larger experiments, including many molecules that would otherwise require terabytes of disk space, quickly consuming the user's storage quota on public Galaxy instances.

### Galaxy tools

The repository containing the developed Galaxy tool wrappers (*qcxms* and *xtb*) is publicly available at https://github.com/RECETOX/galaxytools [20] and archived on Zenodo [19]. All tools are annotated with EDAM identifiers and are registered on bio.tools for better findability [21, 22]. The *xtb molecular optimization* tool uses the BioConda (RRID:SCR_018316) package or the automatically created corresponding biocontainers, depending on the configuration of the respective Galaxy instance [23–25]. The QCxMS tool suite operates within a manually created Docker container containing the QCxMS executable compiled with various compiler optimisations, which are not available when packaging the software through conda. This significantly improves runtime performance and ensures consistency with HPC workflows, leveraging the executable provided via the GitHub repository. The container is based on the following software packages:

- **QCxMS**: Version 5.2.1
- **PlotMS**: Version 6.2.0
- **Python**: Version 3.8.2 (default for the OS, included in Docker container)
- **Operating System**: Linux Ubuntu 20.04 Focal Fossa (base image of the Docker container).



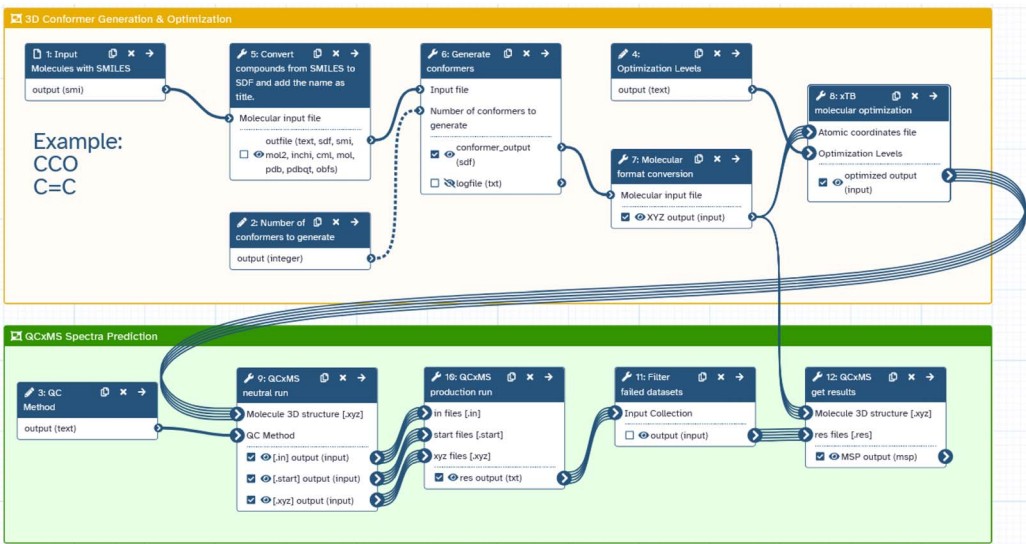

**Figure 1.** Galaxy workflow diagram for EI-MS prediction. The workflow begins with simplified molecular input line entry system (SMILES) strings as input, converts them to a structure data file (SDF) format, and handles files with multiple structures. The SDF file is first processed by the *generate conformers* tool (from the *ChemicalToolbox* [16]) to create multiple conformations for each molecule. Then, the *molecular format conversion* tool converts these into `.xyz` files for subsequent steps. Conformers are then optimised using the *xtb molecular optimization* tool to refine their molecular structure, producing optimised `.xyz` files. The optimised `.xyz` files are then fed into the *qcxms neutral run* tool, which initialises the simulation by generating `.in`, `.start`, and `.xyz` files for each trajectory. These outputs are fed to the *qcxms production run* tool that executes the main simulation and generates detailed `.res` files. Following the simulation, the *qcxms get results* tool processes the `.res` files to generate a standardised `msp` file, consolidating the simulation outcomes. Throughout the workflow, the filter failed jobs tool ensures that only successful job outputs are passed on to subsequent steps, maintaining data integrity and workflow efficiency. Downstream annotation via spectral matching is enabled via the *matchms* Galaxy tools [19].

This approach enhances reproducibility by encapsulating all required software dependencies and configurations [26]. Each step of the workflow is based on an individual Galaxy tool - with all QCxMS-based tools depicted in Figure 2 leveraging the same container - according to best practices recommendations for FAIR computational workflows [27].

### xTB molecular optimization

The *xtb molecular optimization* tool is responsible for optimising molecular structures. The level of accuracy for the geometry optimisation can be adjusted according to user needs. **Input:** The initial input for the *xtb molecular optimization* tool in `xyz` format, containing the initial coordinates of the molecules that need to be optimised. **Output:** Optimised `xyz` file containing the coordinates of the molecules after optimisation. The coordinates have been adjusted to minimise the energy of the molecular structure.

### QCxMS neutral run

The *qcxms neutral run* Galaxy tool initiates quantum chemistry simulations using either the GFN2-xTB or GFN1-xTB SQM methods. The tool performs computations using the QCxMS software, processing the molecule structure twice and utilising a Python script (`rename.py`) to organise and rename the output files into collections while preserving the QCxMS directory structure. **Input:** An `xyz` coordinate file representing the 3D structure of the molecule. **Outputs:** Collections of `.in`, `.start`, and `.xyz` files, containing information about

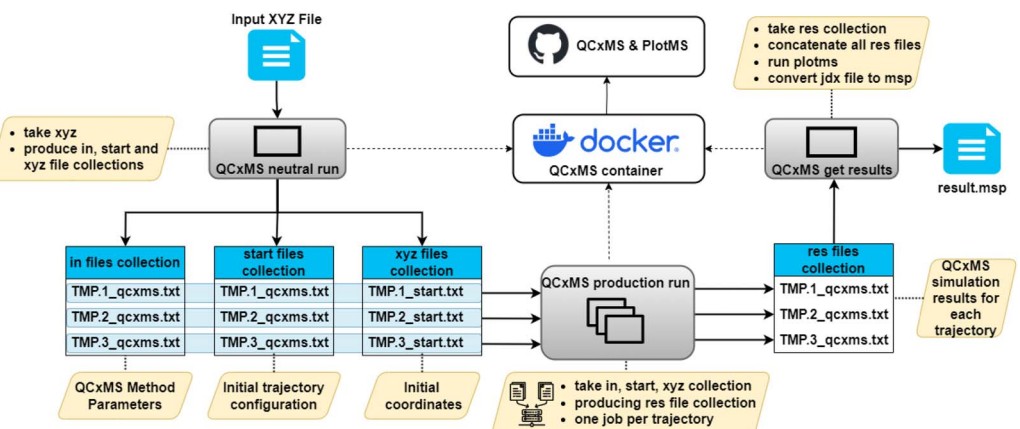

**Figure 2.** Technical overview of the QCxMS Galaxy tool suite. The spectra prediction process starts with an input XYZ file, which undergoes a neutral run in QCxMS to generate method parameters, initial trajectory configurations, and initial coordinates. These files are then used in a production run to simulate various trajectories in parallel and produce `.res` files. The Docker container encapsulates the entire process, ensuring consistency and reproducibility. In the final stage, the results are compiled and analysed, generating a result file, `result.msp`.

the individual trajectories for the production run. Optional extended outputs include `log` files and additional trajectory data.

### QCxMS production run

The *qcxms production run* Galaxy tool processes the trajectories generated by the neutral run and performs detailed quantum chemistry calculations to simulate mass spectra. The tool initiates one job per trajectory, recreates the directory structure (`TMPQCXMS`) using a Python script (`create_folder_structure`) and performs the calculation. **Inputs:** The tool accepts three files, `.in`, `.start`, and `.xyz`, which are obtained from the *qcxms neutral run* tool. **Outputs:** Optional log file (`log`) storing extended execution details. Result files (`res_output` collection) containing output data from the production run.

### QCxMS get results

The *qcxms get results* Galaxy tool processes the `.res` files generated by the production run to produce a simulated mass spectrum (`msp` file). The tool aggregates multiple `.res` files into a single temporary result file (`tmpqcxms.res`) using the Python script (`get_res`), and uses the PlotMS tool for final processing and a shell script (`msp_out.sh`) to produce the final output file in `msp` format. **Inputs:** The tool accepts a collection of `.res` files obtained from the *qcxms production run* tool and the initial `xyz` coordinate file passed in the *qcxms neutral run* tool. **Output:** Predicted high-resolution mass spectra (`msp_output`) for all molecules contained in the starting SDF file with identifiers in `msp` format.

## RESULTS AND DISCUSSION
## Case studies on selected molecules

The integrated workflow was demonstrated using four selected molecules: ethylene (ethene; $C_2H_4$), mirex (1,2,3,4,5,5,6,7,8,9,10,10-dodecachloropentacyclo[5.3.0.0$^{2,6}$.0$^{3,9}$.0$^{4,8}$]decane; $C_{10}Cl_{12}$), benzophenone (diphenylmethanone; $C_{13}H_{10}O$), and enilconazole (1-[2-(2,4-dichlorophenyl)-2-prop-2-enoxyethyl]imidazole; $C_{14}H_{14}Cl_2N_2O$), each with



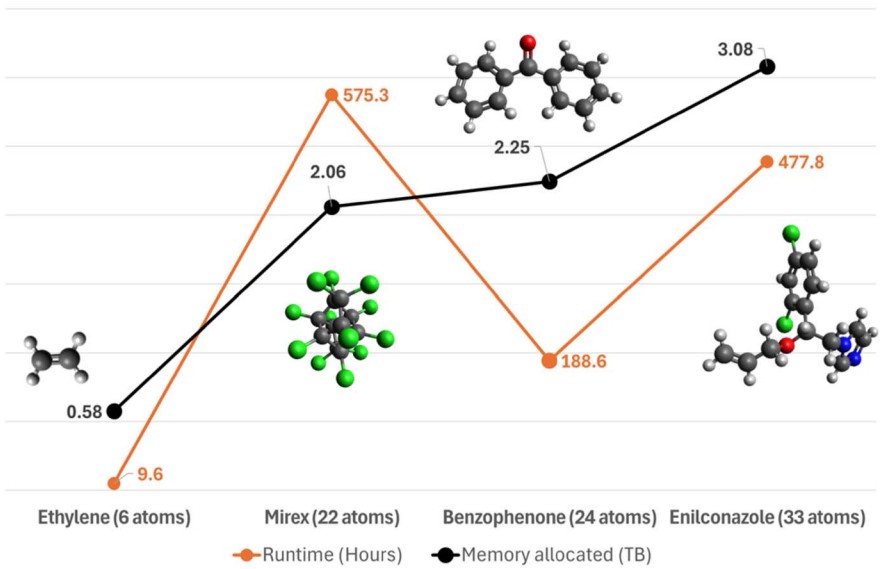

**Figure 3.** Computational performance metrics for molecular simulations using the QCxMS workflow for ethylene (6 atoms), mirex (22 atoms), benzophenone (24 atoms), and enilconazole (33 atoms). The runtime (orange line, in hours) and memory allocated (black line, in terabytes) demonstrate an overall increase in computational demands as the number of atoms increases. Notably, mirex exhibits a higher runtime compared to benzophenone despite having fewer atoms, indicating variations in computational complexity due to its molecular structure.

**Table 1.** Runtime statistics of the experiment obtained using bioblend (RRID:SCR_014557) [30]. All times are given in hours.

|                  | Ethylene | Mirex  | Benzophenone | Enilconazole    |
|------------------|----------|--------|--------------|-----------------|
| Number of atoms  | 6        | 22     | 24           | 33              |
| Atoms            | C, H     | C, Cl  | C, H, O      | C, H, N, O, Cl  |
| CPU cores        | 155      | 555    | 605          | 830             |
| Job Runtime      | 9.62     | 575.26 | 188.62       | 477.84          |
| Memory (TB)      | 0.58     | 2.06   | 2.25         | 3.08            |
| CPU usage time   | 0.65     | 2.75   | 2.83         | 3.89            |
| CPU user time    | 0.48     | 2.09   | 2.12         | 2.84            |
| CPU system time  | 0.17     | 0.67   | 0.71         | 1.05            |

varying chemical and structural complexities. The workflow was run on usegalaxy.eu, the history exported and results deposited on Zenodo as RO-crates [28, 29].

### Scalability and complexity

Simple molecules, such as ethylene with just 6 atoms, were processed quickly, requiring approximately 5 times less central processing unit (CPU) cores and memory, with a job runtime 50 times smaller and 6 times less CPU usage compared to the complex enilconazole molecule with 33 atoms. These parameters are summarised in Table 1, and the relationships between runtime, memory allocation, and number of atoms are illustrated in Figure 3. The job runtimes, memory usage, and CPU demands increased with both molecular complexity and the number of atoms.



### Chemical composition impact

The presence of specific elements, such as chlorine in compounds like mirex and enilconazole, contributed to the computational complexity, resulting in higher resource consumption. For instance, predicting the spectrum of mirex, with 22 atoms including chlorine, took about 3 times longer than that of the comparably sized benzophenone. The details of these computations are shown in Table 1, while the impact of chemical composition on computational resource requirements is visualised in Figure 3. This underscores that the elemental composition of a molecule significantly affects the computational complexity.

### Workflow efficiency

The workflow efficiently leverages available computing resources by implementing data-level parallelism on the Galaxy job submission layer. The fragmentation of each molecule is simulated using a finite number of trajectories (by default 25× the number of atoms). Each trajectory is then simulated independently. The *qcxms production run* tool is executed for each trajectory, and the results are combined later. This is implemented using (nested) collections and batch job submission in Galaxy.

## Workflow integration and training

The workflow's user-friendly interface ensures ease of use, enabling researchers to predict mass spectra covering a wide range of molecular complexities, from simple hydrocarbons to complex organochlorine compounds, without extensive computational knowledge. The provided tools also integrate seamlessly with existing infrastructure for data preparation via the *ChemicalToolbox* and tools for downstream analysis, such as *matchms* for spectral similarity calculation. Comprehensive training published on the Galaxy Training Network (GTN) [11] explains each step of the workflow to give users a better understanding of the tools' functions. It also incorporates a set of features facilitating the learning process and allows users to choose the track of analysis that best meets their needs.

## Accessibility and usability

The workflow presented in this paper is freely accessible on WorkflowHub and can be launched on usegalaxy.eu from the website [31, 32]. Providing a robust tool set on the public Galaxy instances greatly enhances the accessibility of domain-specific expert research software (i.e., QCxMS). Furthermore, this provides access to HPC infrastructure, enabling researchers to conduct advanced MS data analysis regardless of location, institutional resources, or computational expertise. To further increase the accessibility, a tutorial on using the QCxMS tool suite has been published on the GTN [33]. By integrating the FAIR principles [34], the tutorial can be used by both educators and scientists with different computational expertise.

## Implications

- **Wider adoption:** The user-friendly and freely accessible workflow encourages its application across various research fields, including organic chemistry, biochemistry, metabolomics and exposomics.
- **Collaborative research:** The incorporation of tools into the Galaxy platform supports the efficient sharing of workflows and results, thereby fostering collaborative research.

- **Enhanced efficiency:** The integration accelerates scientific discovery by making QC-based spectra predictions available to the scientific community and by improving the efficiency of updating QC-based methods for fragmentation prediction. This enhancement helps streamline the process of MS annotations.

## Future work

The QCxMS tools and workflows could be enhanced by integrating advanced *ab initio* quantum mechanical methods, such as density functional theory for trajectory calculations, provided that permissive licenses and computational resources are available [35]. Additional SQM methods, such as the OMx family, which have been applied in other publications [36–38], could also be added. Going beyond EI mass spectra, QCxMS can also be used to predict spectra obtained from collision induced dissociation, and the addition of this operational mode would further increase the range of possible applications.

## CONCLUSIONS

A complete Galaxy workflow for QC-based EI mass spectra prediction has been developed and released in the WorkflowHub repository [39], building upon previous semi-automated HPC workflows [14]. This integration, available on usegalaxy.eu, is complemented by comprehensive Galaxy training, ensuring researchers can effectively utilise these advanced tools.

## AVAILABILITY OF SOURCE CODE AND REQUIREMENTS

### QCxMS galaxy tools

- Project name: QCxMS Galaxy Tools
- Project home page: https://github.com/RECETOX/galaxytools/tree/master/tools/qcxms
- Operating system(s): NA
- Programming language: Fortran, Python, Bash
- Dockerfile: https://github.com/RECETOX/qcxms-dockerfile
- Container: https://hub.docker.com/r/recetox/qcxms-docker
- WorkflowHub: https://doi.org/10.48546/WORKFLOWHUB.WORKFLOW.897.3
- Dockstore: https://doi.org/10.5281/zenodo.15303974
- License: MIT
- Software Heritage ID:
  swh:1:snp:185690f3e21f005fd085a1bf9400627de8c84b59;origin=https://github.com/RECETOX/galaxytools

### QCxMS

- Project name: QCxMS
- Project home page: https://github.com/qcxms/QCxMS
- Operating system(s): Linux, OSX
- Programming language: Fortran, Bash
- Dockerfile: https://github.com/RECETOX/qcxms-dockerfile
- Container: https://hub.docker.com/r/recetox/qcxms-docker
- License: LGPL-3.0
- bio.tools: https://bio.tools/qcxms

- Software Heritage ID:
  swh:1:snp:d8d2934bea7b8490c49acaff115332fa15b8a5a7;origin=https://github.com/qcxms/QCxMS
- RRID: SCR_026928

### xTB
- Project name: xTB
- Project home page: https://github.com/grimme-lab/xtb
- Operating system(s): Linux, OSX
- Programming language: Fortran
- License: LGPL-3.0
- bio.tools: https://bio.tools/xtb_molecular_optimization
- Software Heritage ID:
  swh:1:snp:f9b8d28ac5ee4199b06f6fe9c6ca9bc50cb07ab8;origin=https://github.com/grimme-lab/xtb
- RRID: SCR_026929

## DATA AVAILABILITY
### GitHub repositories
Both the QCxMS (https://github.com/qcxms/QCxMS) and xTB
(https://github.com/grimme-lab/xtb) software packages are developed open source on
GitHub. The QCxMS and xTB tool suites are part of the collection of Galaxy tool wrappers
developed by the RECETOX RI at https://github.com/RECETOX/galaxytools. Versions of
record snapshot of these GitHub repositories have been archived in the Software Heritage
Library [40].

### Zenodo data files
Archives of the galaxytools repository are available on Zenodo [19] and the Software
Heritage [40]. The Galaxy histories containing the workflow executions for the four
example molecules are archived on Zenodo [29].

### Resource identifiers
The workflow presented in this manuscript is archived on WorkflowHub [31]. Both the
QCxMS (https://bio.tools/qcxms) and xTB (https://bio.tools/xtb_molecular_optimization)
packages have been indexed on bio.tools for improved findability. The resources have also
been registered with SciCrunch and given the following RRIDs: SCR_026928 (QCxMS) and
SCR_026929 (xTB). The GTN training is available at https://gxy.io/GTN:T00458 [33].

### LIST OF ABBREVIATIONS
CPU, central processing unit; EI, electron ionization; FAIR, Findability, Accessibility,
Interoperability, Reusability; GTN, Galaxy Training Network; HPC, high-performance
computing; MS, mass spectrometry; QC, quantum chemistry; SDF, structure data file;
SMILES, simplified molecular input line entry system; SQM, semi-empirical quantum
mechanics; xTB, extended tight binding.

## DECLARATIONS

### Consent for publication

Not applicable.

### Competing interests

The authors declare that they have no competing interests.

### Authors' contributions

WYR: Analysis, software, visualization, writing – original draft, review and editing; ZA: Software, visualization, writing – review and editing; JJ: Visualization, validation, writing – review and editing; HH: Analysis, software, visualization, supervision, writing – original draft, review and editing; JK: Funding acquisition, writing – review and editing; EJP: Resources, supervision, writing – review and editing.

### Funding

The work was financially supported by the project SALVAGE (CZ.02.01.01/00/22_008/0004644), financed by MEYS – Co-funded by the European Union. The authors thank the RECETOX Research Infrastructure (LM2023069), financed by the Ministry of Education, Youth and Sports (MEYS), and the Operational Programme Research, Development and Education (the CETOCOEN EXCELLENCE project No. CZ.02.1.01/0.0/0.0/17_043/0009632) for supportive background. This work was supported by the European Union's Horizon 2020 research and innovation programme under grant agreement No 857560 (CETOCOEN Excellence). This publication reflects only the author's view and the European Commission is not responsible for any use that may be made of the information it contains.

### Acknowledgements

The authors acknowledge the support of the Freiburg Galaxy Team and Björn Grüning, Bioinformatics, University of Freiburg (Germany), funded by the German Federal Ministry of Education and Research BMBF grant 031 A538A de.NBI-RBC and the Ministry of Science, Research and the Arts Baden-Württemberg (MWK) within the framework of LIBIS/de.NBI Freiburg.

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
