## [Reviewer Report]

Indicate in the comments box below whether you are happy with the changes made or if the manuscript is unacceptable.Comments on revised manuscriptAll points have been adequately addressed, and the manuscript is now suitable for publication. Regarding the tinit / etemp issue: etemp should not be overwritten by tinit, as they are independent parameters — tinit reflects the actual temperature, while etemp is the electronic temperature used to account for missing static correlation effects of the respective electronic structure method. The documented default value of 5000 K for etemp is correct. As previously noted, this issue does not occur with the standalone version. Although somewhat puzzling, the bug has now been resolved in the Galaxy tool workflow, and my test calculations produced the expected results.Indicate in the comments box below whether you are happy with the changes made or if the manuscript is unacceptable.Comments on revised manuscriptAll points have been adequately addressed, and the manuscript is now suitable for publication. Regarding the tinit / etemp issue: etemp should not be overwritten by tinit, as they are independent parameters — tinit reflects the actual temperature, while etemp is the electronic temperature used to account for missing static correlation effects of the respective electronic structure method. The documented default value of 5000 K for etemp is correct. As previously noted, this issue does not occur with the standalone version. Although somewhat puzzling, the bug has now been resolved in the Galaxy tool workflow, and my test calculations produced the expected results.

---

## [Editor Report]

Editor’s AssessmentStructural annotation of mass spectrometry (MS) data is a critical but challenging step in many scientific workflows. Quantum chemistry (QC)-based predictions requiring even more challenging High-performance computing (HPC) environments that researchers with limited computational knowledge struggle to utilise. This paper presents a robust workflow that leverages interoperable file formats for molecular structures to ensure seamless integration, and makes it easier for users with command-line experience to perform semi-empirical quantum mechanical (SQM)-based predictions using advanced computational tools without needing in-depth expertise. Galaxy QCxM uses the popular Galaxy workflow system, compatible with other chemo-informatics tools from the Galaxy ChemicalToolbox, and leverages standard data formats supported by Open Babel. The workflow is freely accessible on WorkflowHub and can be launched on the usegalaxy.eu Galaxy HPC server, and is complemented by Galaxy training as well. The workflow for four molecules are included as examples, and the peer reviewers managed to test these ( flagging a minor bug that was fixed). Going forward QCxMS could also be used to predict spectra obtained from collision induced dissociation (CID) and the addition of this operational mode would further increase the range of possible applications.Editor’s AssessmentStructural annotation of mass spectrometry (MS) data is a critical but challenging step in many scientific workflows. Quantum chemistry (QC)-based predictions requiring even more challenging High-performance computing (HPC) environments that researchers with limited computational knowledge struggle to utilise. This paper presents a robust workflow that leverages interoperable file formats for molecular structures to ensure seamless integration, and makes it easier for users with command-line experience to perform semi-empirical quantum mechanical (SQM)-based predictions using advanced computational tools without needing in-depth expertise. Galaxy QCxM uses the popular Galaxy workflow system, compatible with other chemo-informatics tools from the Galaxy ChemicalToolbox, and leverages standard data formats supported by Open Babel. The workflow is freely accessible on WorkflowHub and can be launched on the usegalaxy.eu Galaxy HPC server, and is complemented by Galaxy training as well. The workflow for four molecules are included as examples, and the peer reviewers managed to test these ( flagging a minor bug that was fixed). Going forward QCxMS could also be used to predict spectra obtained from collision induced dissociation (CID) and the addition of this operational mode would further increase the range of possible applications.

---

## [Reviewer Report]

Reviewer name and names of any other individual's who aided in reviewerGareth PriceDo you understand and agree to our policy of having open and named reviews, and having your review included with the published manuscript. (If no, please inform the editor that you cannot review this manuscript.)YesIs the language of sufficient quality?YesPlease add additional comments on language quality to clarify if neededIs there a clear statement of need explaining what problems the software is designed to solve and who the target audience is? YesAdditional CommentsIs the source code available, and has an appropriate Open Source Initiative license <a href="https://opensource.org/licenses" target="_blank">(https://opensource.org/licenses)</a> been assigned to the code?YesAdditional CommentsAs Open Source Software are there guidelines on how to contribute, report issues or seek support on the code?YesAdditional CommentsIs the code executable?YesAdditional CommentsIs installation/deployment sufficiently outlined in the paper and documentation, and does it proceed as outlined?YesAdditional CommentsIs the documentation provided clear and user friendly?YesAdditional CommentsIs there enough clear information in the documentation to install, run and test this tool, including information on where to seek help if required?Additional CommentsIs there a clearly-stated list of dependencies, and is the core functionality of the software documented to a satisfactory level?YesAdditional CommentsHave any claims of performance been sufficiently tested and compared to other commonly-used packages? YesAdditional CommentsPerformance was measured internally on Galaxy. This paper is not about a comparison of performance in different hardware configurations.Is test data available, either included with the submission or openly available via cited third party sources (e.g. accession numbers, data DOIs)?YesAdditional CommentsAre there (ideally real world) examples demonstrating use of the software? YesAdditional CommentsIs automated testing used or are there manual steps described so that the functionality of the software can be verified?YesAdditional CommentsAny Additional Overall Comments to the AuthorThis paper outlines and describes a valuable activity in the analytical space, expanding the audience for analytical methodologies that are restricted by a high knowledge barrier. The use of Galaxy and the associated Galaxy Training Network is an efficient method to expose workflows that are historically confined to only HPC environments. I recommend this paper for publication, with minor revisions. Page 5 – the description of “QCxMS Neutral Run” and “QCxMS Production Run” and “QCxMS Get Results” all refer to python scripts. It is unclear if these scripts have been wrapped and made available as Galaxy tools that can be used within Galaxy workflows. I strongly suspect this is the case, they are available as Galaxy tools, but more clarity would be good. The impression should not be that users are required to trigger python scripts but rather that the workflow can be run in entirety in Galaxy. Page 6 – “Scalability and Complexity: Simple molecules, such as ethylene with just 6 atoms, were processed quickly, requiring approximately 5 times less slots and memory, with a job runtime 50 times smaller and 6 times less central processing unit (CPU) usage compared to the complex enilconazolemolecule with 33 atoms.” The language of slots and CPUs should be corrected to be consistent or a clear definition of slots and CPU provided to justify the two terms. The same applies to Table 1 Page 6 - Chemical Composition Impact: The presence of specific elements, such as chlorine in compounds like mirex and enilconazole, contributed to the computational complexity, resulting in higher resource consumption. For instance, predicting the spectrum of mirex, with 22 atoms including chlorine, took about 3 times as long as the comparably sized benzophenone. The details of these computations are shown in Table 1, while the impact of chemical composition on performance is visualised in Figure 3. This underscores that the elemental composition of a molecule significantly affects performance. The word performance needs clarification or definition. Is it a measure of performance associated with run time, model accuracy or an aggregate of measures? Comment - within the text individual tools are mentioned, such as matchms. As tools are core to Galaxy (as is their code to HPC), I suggest they could be italicised to stand out in text more easily.RecommendationMinor Revisions

---

## [Reviewer Report]

Upload additional filesTRR-202410-01R01/stage_files/TRR-202410-01/Review MS/neutral_run_out.texReviewer name and names of any other individual's who aided in reviewerStefan GrimmeDo you understand and agree to our policy of having open and named reviews, and having your review included with the published manuscript. (If no, please inform the editor that you cannot review this manuscript.)YesIs the language of sufficient quality?YesPlease add additional comments on language quality to clarify if neededIs there a clear statement of need explaining what problems the software is designed to solve and who the target audience is? YesAdditional CommentsIs the source code available, and has an appropriate Open Source Initiative license <a href="https://opensource.org/licenses" target="_blank">(https://opensource.org/licenses)</a> been assigned to the code?YesAdditional CommentsAs Open Source Software are there guidelines on how to contribute, report issues or seek support on the code?YesAdditional CommentsIs the code executable?YesAdditional CommentsIs installation/deployment sufficiently outlined in the paper and documentation, and does it proceed as outlined?YesAdditional CommentsIs the documentation provided clear and user friendly?YesAdditional CommentsIs there enough clear information in the documentation to install, run and test this tool, including information on where to seek help if required?YesAdditional CommentsIs there a clearly-stated list of dependencies, and is the core functionality of the software documented to a satisfactory level?YesAdditional CommentsHave any claims of performance been sufficiently tested and compared to other commonly-used packages? YesAdditional CommentsIs test data available, either included with the submission or openly available via cited third party sources (e.g. accession numbers, data DOIs)?YesAdditional CommentsAre there (ideally real world) examples demonstrating use of the software? YesAdditional CommentsIs automated testing used or are there manual steps described so that the functionality of the software can be verified?NoAdditional CommentsIncluding a test example for a small molecule would be beneficial. While automated testing (e.g., via unit tests) may be challenging to implement for this program, it would be highly valuable. The existing test for the lindane molecule in the GitHub repository does not seem particularly useful, as the number of trajectories is too low. With such a small number of trajectories, the results for the spectrum will vary significantly between different runs, making it less reliable as a test case.Any Additional Overall Comments to the AuthorThe authors present an open-source workflow for using the EI-MS mode of QCxMS with GFN2-xTB or GFN1-xTB via a graphical user interface on the freely accessible Galaxy HPC platform. The work is well done and clearly presented in the manuscript. It has the potential to make the QCxMS program accessible to a broader scientific community, particularly for researchers who are not experienced with Linux server environments, and should therefore be published. However, I encountered an issue when testing the workflow on usegalaxy.eu. I performed test calculations for methanol and pentanone, but the workflow did not function as expected and did not generate an .msp file. In the neutral run, the molecules were completely fragmented, likely due to the default initial temperature (tinit) being set to 5000K instead of the QCxMS default of 500K. This temperature is far too high and leads to unphysical results. Interestingly, the input file correctly specifies 500K, suggesting a potential bug in how the parameters are handled within the workflow. Notably, this issue does not occur in the standalone QCxMS program. To ensure meaningful results, this bug should be addressed. For reference, I have attached the log/output files from the QCxMS neutral run and the produced qcxms.in file, which seems fine to me and works with the QCxMS standalone, which sets the initial temperature correctly to 500 K. Some minor remarks. 1.) As I did a test calculation, the tmax setting for the MDs was set to 20ps. This is very expensive and in almost no case necessary, I would set the default to 5ps to save computational resources. 2.) Overall, I managed to get the calculation working, but it was very it was very tedious for me. More hints what files should be selected for each step and input field would be helpful. Maybe it would be helpful to give a complete spectra calculation (neutral run, production run and get results) a name and automatically select the produced files then for the next step or even better integrate everything into one step.RecommendationMajor Revisions